# Bedside Teaching in Rural Family Medicine Education in Japan

**DOI:** 10.3390/ijerph19116807

**Published:** 2022-06-02

**Authors:** Ryuichi Ohta, Chiaki Sano

**Affiliations:** 1Community Care, Unnan City Hospital, 96-1 Iida, Daito-cho, Unnan 699-1221, Japan; 2Department of Community Medicine Management, Faculty of Medicine, Shimane University, 89-1 Enya cho, Izumo 693-8501, Japan; sanochi@med.shimane-u.ac.jp

**Keywords:** bedside teaching, rural family education, medical resident, medical teacher, nurses, community hospitals, Japan

## Abstract

Bedside teaching is essential in family medicine education so that residents may learn about various clinical conditions and develop professional skills. In particular, bedside teaching is useful in a rural context because rural family medicine deals with a broad scope of biopsychosocial problems among older patients. Accordingly, based on an inductive thematic analysis, we propose a framework for bedside teaching in rural family medicine education, which consists of four themes: accommodation of different learners, near-peer learning, the change in engagement of medical teachers in bedside teaching, and driving interpersonal collaboration. Bedside teaching can promote interactions between different medical learners. Near-peer learning in bedside teaching compensates for the limited availability of educators and improves learners’ motivation for self-directed learning. Through bedside teaching, medical learners can observe each other and provide constructive feedback, thereby improving their relationships and learning. For effective bedside teaching, medical educators should facilitate learners and collaborate with other medical professionals. Additionally, bedside teaching should accommodate a variety of learners, facilitate near-peer and self-directed learning, educators’ involvement based on cognitive apprenticeship, along with interprofessional collaboration with nurses. Interprofessional collaboration between rural family medicine teachers, learners, and nurses may improve the quality of patient care due to the increased understanding between patients and other medical staff in hospitals.

## 1. Background

Bedside teaching is essential for family medicine education so as to educate residents about various conditions [1]. Family medicine deals with many kinds of medical issues faced by patients, often with biopsychosocial aspects. Physicians who specialize in family medicine are called family physicians and work in various clinical settings, such as hospitals, clinics, and public health centers; therefore, family physicians have to learn ways to solve various healthcare issues in their training [1]. In family medicine education, various educational methods are used, such as case-based learning, problem-based learning, and on-the-job training. One type of on-the-job training is bedside teaching [2].

Bedside teaching is performed in inpatient clinical wards to teach medical knowledge, skills, and professionalism. Usually, medical teachers and students go to patients’ bedsides and examine the patients together. Medical teachers observe their students’ performance and provide feedback in front of the patients or outside their rooms [2,3]. Bedside teaching can be performed in various clinical contexts, such as morning rounds, formal educational rounds, and attending rounds, to assess learners’ management of patients [3,4], each of which can improve learning among medical students. Nowadays, the prevalence of bedside teaching has declined because of our reliance on computer-based working and laboratory tests for the management of patient conditions [5,6]. Nevertheless, to sustain effective, high-quality education, bedside teaching should be encouraged and improved [7].

In particular, bedside teaching is useful in the rural context because rural family medicine deals with a broad range of biopsychosocial problems among older patients, especially in developed countries such as Japan. As young people move from rural communities to urban areas to find jobs, and rural community hospitals lack sufficient medical professionals devoted to full-time care, family medicine educators often treat patients in these rural communities as well as providing education [8,9,10,11]. When medicine educators become too busy with clinical practice, they may not be able to devote enough time to residents [12,13].

Furthermore, certain characteristics of rural contexts can affect family medicine education: due to the lack of rural medical education, patients in rural hospitals may not be used to being examined by medical residents and teachers [9]. Other medical professionals in rural hospitals, such as nurses, may not have experienced educational situations where medical students and teachers discuss patients’ conditions in medical wards [14,15]. Therefore, the application of bedside teaching in rural family medicine education can be challenging. At present, there is a lack of evidence regarding the most effective ways of conducting bedside teaching in rural contexts [16]. Nonetheless, bedside teaching is necessary for effective rural family medicine education. Therefore, this study’s purpose was to construct an effective framework for the application of bedside teaching in rural family medicine education.

## 2. Materials and Methods

This qualitative research was conducted to clarify how bedside teaching in rural family medicine education was being performed and how it could be improved by proposing a teaching framework. Ethnographic research and interviews were conducted from 1 April 2020 to 31 December 2021.

### 2.1. Setting

The study setting was Unnan City Hospital, located in southeast Shimane prefecture in rural Japan. At the time of the study, the hospital had 281 care beds, of which 160 were acute care, 43 comprehensive care, 30 rehabilitation, and 48 chronic care. The nurse-to-patient ratios were 1:10 for acute care, 1:13 for comprehensive care, 1:15 for rehabilitation, and 1:25 for chronic care. The hospital had 27 physicians, 197 nurses, 7 pharmacists, 15 clinical technicians, 37 therapists, 4 nutritionists, and 34 clerks.

The hospital followed a rural family medicine education curriculum, which included three family medicine teachers. Under this curriculum, residents experienced various clinical situations in the treatment of their patients. In their first year, residents worked at the Unnan City Hospital and treated typical diseases in both inpatient and outpatient situations. In the following year, residents worked at a rural clinic (Kakeya Clinic) for 6 months to learn home care and community-oriented primary care. To broaden their scope of practice in internal medicine, pediatrics, and emergency medicine, residents also worked at a general or community hospital for 18 months.

Each clinical setting included a medical teacher. The curriculum could be utilized to educate a maximum of three residents per teacher, simultaneously. One resident in the years 2018 and 2019, and three in the years 2020 and 2021, engaged in the curriculum.

Medical students and junior residents had received rural family medicine education at medical universities and tertiary hospitals. They trained in family medicine at the rural hospital for two weeks to a month with medical teachers and family medicine residents. It was mandatory for them to work at the rural hospital as part of their university or hospital curriculum. Each year, the rural hospital accommodates 40 to 50 medical students and junior residents for training.

### 2.2. Participants

The participants represented all stakeholder groups involved in rural family medicine curriculum: medical students, medical residents, medical teachers, and nurses, who collaborated with the students and residents in the treatment of patients from April 2019 to March 2021. We used purposive sampling to address the research purposes for ethnography and semi-structured interviews.

### 2.3. Ethnography and Semi-Structured Interviews

The first researcher conducted ethnography and semi-structured interviews with the participants: medical students, residents, teachers, and patients. In the ethnography study, participants were medical students, residents, teachers, nurses, and patients in bedside teaching situations. This researcher’s specialties are family medicine, medical education, and public health. The researcher worked in all hospital wards, observed interactions among medical students, residents, teachers, and nurses at bedsides, and took field notes during this process. During the observation period, the researcher interviewed all residents, medical teachers, and nurses. The interview guide included four questions: What did you think of bedside teaching? How did you evaluate the effectiveness of bedside teaching? How do you evaluate difficulty in bedside teaching? Do you have any ideas on how to improve the quality of bedside teaching in family medicine? Each interview lasted about 30 min and was recorded and transcribed verbatim. The transcript was shared with the interviewee to confirm the credibility of the content.

### 2.4. Analysis

Inductive thematic analysis was used. After reading the field notes and interview transcriptions in depth, the first researcher coded the contents and developed codebooks based on repeated reading of the research materials, as initial coding for reliability. This study used process and concept coding, where the researcher induced, merged, deleted, and refined the concepts and themes by going back and forth between the research materials and the initial coding. For triangulation, the concepts and themes were discussed between the first and second researchers. Each month, the interview data were iteratively collected from different participants during the research period for theoretical saturation. The tentative analysis results were also shared with the participants for audit trial. Finally, the theory was discussed by both researchers, who ultimately reached agreement on the final themes. Our research was approved by the Unnan City Hospital ethical committee (Approval no. 20190005).

## 3. Results

In total, 64 medical students (46.9% female), 63 medical residents (33.3% female), 6 family medicine residents (33.3% female), and 22 nurses (all female) were observed and interviewed in the research period. Based on the thematic analysis, the framework of bedside teaching in rural family medicine education comprises four themes: accommodation of different learners, near-peer learning, the change in the engagement of medical teachers in bedside teaching, and driving interpersonal collaboration. Effective bedside teaching requires the integration of these four themes (Figure 1).

Bedside teaching can promote interactions between different medical learners. Near-peer learning is essential in bedside teaching to compensate for a shortage of physicians and to improve learners’ motivation for self-directed learning. Through bedside teaching, medical learners can observe each other and provide constructive feedback about their learning, which may improve their relationships and promote better learning. The change in engagement of medical teachers in bedside teaching can be vital for self-directed and near-peer learning. For effective bedside teaching, medical educators should therefore facilitate learners and communicate and collaborate with other medical professionals.

### 3.1. Accommodation of Different Learners

Bedside teaching promotes interactions between different medical learners. The involvement of different medical learners, such as medical students, residents, and nurse practitioners, can provide different perspectives for patients [14]. The participating medical students were interested in the pathophysiology of patients’ conditions based on their own education at medical universities, while residents tended to be interested in the management of patients’ conditions, and nurse practitioners were interested in the gap between medical care in rural hospitals and patients’ lives at home. As one of the medical students stated, “*I am not used to considering clinical decisions in real cases, but I could learn a lot here about the pathophysiology of diseases fitting my learning in the university*.” One resident stated: “*In the rural hospital, I got to experience various cases among older patients. I could apply my medical knowledge and skills to treat them.*” Another resident stated: “*We were able to gain various perspectives through our interaction with medical students and teachers. They have different perspectives in medicine. Their opinions promoted a deeper consideration of patient care.*” Through discussion, learners can acquire different perspectives about patient care, which can improve the quality of patient care in rural hospitals [17].

For effective bedside teaching, the preparedness of medical learners is essential. Therefore, bedside teaching participants needed preparation and motivation [18]. One of the medical students stated: “*In the medical university, I never experienced clinical rounds in wards with residents and medical teachers. So, I could not grasp bedside teaching at first.*” Another medical resident stated: “*Bedside teaching is interesting. Initially, I was embarrassed by being observed by medical teachers and medical students in front of various patients.*” Medical teachers had to instruct medical learners on bedside teaching before participating in rural family medicine education [19]. When learners are interested in bedside teaching-based learning, they participate in rural family medicine education. One of the students stated: “*Family medicine may be popular among a specific group of medical students and residents. This curriculum should be elective.*” Based on the SPICES model in community-based medical education, rural family medicine education can have the elective component provided to prepared learners [20].

### 3.2. Near-Peer Learning

Near-peer learning is essential for bedside teaching, compensating for a shortage of medical teachers and improving learners’ motivation for self-directed learning. Near-peer learning refers to students instructing other students, evaluating the outcomes of their learning, and providing assessments and feedback [21]. In bedside teaching, participants discuss their own perspectives from different contexts. Through this discussion, in addition to learning patient care, the participants can understand different contexts of learning and build effective relationships with other participants. One of the medical residents stated: “*In the discussion of the patients’ condition, I could share my ideas with other residents, medical teachers, and students. The sharing process enabled me to learn others’ perspectives because they responded to my opinions.*” One of the students added: “*I could present my ideas to others during the bedside teaching. Initially, I hesitated, but later could understand others through continual discussion and dialogue.*” Learners’ understanding of each other can drive their learning because they provide each other with feedback in the method appropriate to each participant based on near-peer learning [22].

Through bedside teaching, medical learners observe each other and provide constructive feedback about their learning based on their relationships. This feedback involves learning about professionalism and interprofessional collaboration. One of the residents stated: “*During the bedside teaching, medical teachers show professionalism in medicine and interprofessional collaboration. During the teaching, teachers involved various learners in the discussion of professionalism and interprofessional collaboration. This teaching can be beneficial.*” The participants shared their knowledge and skills and discussed how to apply them to the present context. One of the students stated: “*I was able to show my abilities to the other members and got feedback from them. Their feedback will be important in bedside learning.*” This process motivates the participants to improve their self-directed learning [23,24]. In bedside teaching, sufficient support from teachers can encourage near-peer learning and self-directed learning.

### 3.3. The Change in Engagement of Medical Teachers in Bedside Teaching

The change in the engagement of medical teachers in bedside teaching is vital for self-directed and near-peer learning. Initially, bedside learning needed support from teachers, including guidance on making rounds of the wards, showing professionalism when dealing with patients, and discussing patients’ conditions. As one of the students said, “*We did not get used to bedside teaching. Initially, I was anxious to participate in the teaching.*” Most learners were used to bedside learning and needed to follow the instructions of teachers in clinical wards in a process of cognitive apprenticeship, a term which refers to how learners start by observing clinical practitioners and are gradually given more tasks to perform as their competence grows [25]. One resident confirmed this: “*Medical teachers’ instruction was essential. Their instruction and facilitation could help me to understand concretely what and how I could learn in the bedside teaching, such as history taking and physical examinations.*”

In addition, teachers should promote collaboration between participants in bedside teaching by facilitating their relationships. Initially, participants are not used to bedside teaching and fail to establish effective relationships. One of the residents said: “*I did not meet medical students and other residents. I was embarrassed to interact with others at bedsides.*” Teachers had to facilitate discussions by asking participants simple questions about patients’ conditions [16,23]. They had to be careful not to blame the participants and instead respond positively to them. One of the medical students stated: “*Medical teachers teach me how to act in bedside teaching and collaborate with others. For example, I could share my ideas with other students and residents. The circumstances were good and nobody found fault with what I asked them during bedside teaching.*” The participants shared their ideas about patients and were motivated to participate in active bedside teaching sessions [22]. Additionally, teachers needed to facilitate participants’ collaboration in solving patients’ problems by using small group discussions, to motivate self-directed learning in participants and encourage them to share their ideas freely [26]. This process was able to drive near-peer learning. One of the residents stated: “*Whenever I had questions, I could freely ask the medical teachers. They enabled me to solve the questions with other team members. Their attitude motivated me to learn more and collaborate with others.*”

For participants’ learning to progress, teachers’ involvement in bedside teaching needed to be reduced. The participants became accustomed to learning through bedside teaching and were motivated to make rounds and actively discuss patients’ conditions. Teachers’ support then needed to be reduced to facilitate participants’ self-directed and near-peer learning. One of the medical residents stated: “*After getting used to the bedside learning, I could learn a lot with my team members with some amount of help from teachers, though they consciously reduced their involvement in bedside teaching.*” Teachers observed the participants’ interactions and decided how much they should facilitate them. The gradual reduction in the teachers’ involvement in bedside teaching was beneficial for participants’ learning and was an effective usage of rural teachers’ time, which compensated for the shortage of educators. One of the teachers said: “*I could reduce my time in bedside teaching after medical students and residents became self-directed in their learning and could use it effectively in clinical issues.*”

### 3.4. Driving Interpersonal Collaboration

For effective bedside teaching, educators need to communicate and collaborate with other medical professionals, which can drive interprofessional collaboration in rural family medicine. One of the teachers stated: “*Other medical professionals were anxious about the bedside teaching because they were used to the [current] bedside teaching culture and anxious about the possibility of harmful results of bedside teaching.*” Nurses’ understanding of bedside teaching is essential because rural nurses care directly for patients and understand their unique characters [17]. In bedside teaching, medical teams often encounter various patients who require special communication methods because of delirium and problems such as hearing and eyesight impairments. One of the medical residents reported that “*The approaches to patients with various difficulties are challenging. Nurses could help us to approach them with some tips to communicate with them.*” Nurses’ suggestions improved communication and medical teams’ understanding of patients and their needs.

Appropriate timing of nurses’ consultations with medical teams is particularly important. In rural family medicine, many patients experience multiple changes in their clinical courses, and nurses need to discuss these changes with physicians. The interruption of bedside teaching by discussions with nurses could impinge on the participants’ learning [27,28]. One of the medical teachers noted that “*Nurses’ questions interrupting bedside teaching hindered effectiveness. I had to advise the nurse to pay attention to the timing of her questions.*” Nurses need to choose appropriate times for consultation with family medicine teams between patient rounds. Effective consultation can then allow the timely management of patients’ conditions and improve their quality of care in rural family medicine.

## 4. Discussion

This report suggests methods for bedside teaching in rural family medicine. For the effective provision of bedside teaching, near-peer learning should be respected in clinical rounds. Initially, cognitive apprenticeship, continual feedback, and facilitation are needed to drive near-peer and self-directed learning. Reducing the involvement of teachers and encouraging an appropriate understanding among nurses may improve participants’ learning based on apprenticeship and self-directed learning.

For bedside teaching, near-peer learning is essential. In this research, learning shared among medical students and residents improved their knowledge and skills, leading to effective collaborations. Near-peer learning was used for the education of residents in various specialties [21,29]. This kind of learning is effective for improving self-directed learning, which can in turn improve knowledge and skills [30]. In near-peer learning, the involvement of various learners could improve participants’ understanding of different perspectives and help them reflect on their practices [31,32]. As this study shows, near-peer learning helped participants learn by themselves and realize their growth. Medical teachers also realized that they could use their time more effectively for the education and clinical issues of their patients.

In addition, near-peer learning could effectively stimulate both medical students and teachers’ learning because the collaboration among medical learners and teachers could address clinical questions from different perspectives. For effective collaboration among medical learners and teachers, the inclusion of people with different perspectives and backgrounds can be useful [31,32]. Bedside teaching involving various participants based on near-peer learning should therefore be encouraged for rural family medicine education.

Information and communication technology (ICT) can be used to facilitate participation by various professionals. This study was not able to involve various professionals such as therapists, nutritionists, and pharmacists because of their time constraints. Rural healthcare professionals are busy and may not have enough time to attend to patients’ bedsides. Through ICT tools, various professionals could communicate in clinical rounds and efficiently discuss treatment during bedside visits [33]. Future studies could investigate the effective usage of ICT for bedside teaching in community hospitals.

Professionalism should be enhanced by cognitive apprenticeship in bedside teaching. Professionalism is vital for learners to become true physicians but is difficult to teach [34,35]. Direct observations of learners and feedback on their behavior promotes the effective teaching of professionalism [2]. As this study shows, through bedside teaching, medical teachers could observe medical learners on site and suggest better ways of communication. In addition, this study shows that bedside teaching enhanced the relationship between medical learners and teachers, making it easier for them to observe each other and, if necessary, modify their clinical attitudes towards patients and other professionals to improve patients’ care. Bedside teaching is one of the best opportunities to observe learners and provide feedback. By reflecting on actions together, education on professionalism can be effectively promoted [36]. As this article shows, collaboration and mutual observation among medical learners and teachers could improve their practice as medical professionals. Bedside teaching can also improve decision-making for patient management, leading to the improvement of patient care by the family medicine team [36]. Thus, rural family medicine education should use bedside teaching for the effective teaching and usage of medical teachers’ time.

The involvement of nurses in rural family medicine education is critical to improve the quality of bedside teaching. As this article shows, nurses’ suggestions and dialogue with medical learners and teachers improve their professionalism and patient care. For effective learning from nurses, relationships between medical learners, teachers, and nurses need to be established [37]. Nurses at rural hospitals are good mentors, providing strong examples of professionalism and interprofessional collaboration and facilitating the transition of medical learners into practitioners [37]. Medical learners learn professionalism and interprofessional collaboration through discussions with nurses in clinical situations. As this article shows, rural nurses observe medical learners’ and teachers’ practices and discuss them with the learners and teachers. In bedside teaching, residents have to discuss their patients’ condition with nurses and modify their working style from previous clinical styles. Rural nurses can teach rural hospital culture and alleviate residents’ work difficulties [37]. Rural nurses can observe learners in clinical situations, including bedside teaching, and give constructive feedback to both teachers and learners. However, this research shows that application of bedside teaching might confuse nurses and patients in clinical wards. To improve effective bedside teaching, the collaboration with nurses should be enhanced, and the method of bedside teaching should be discussed with nurses. Involving nurses in education can also facilitate rural bedside teaching among teachers and residents with regard to professionalism and interprofessional collaboration.

This study has limitations. The first issue is the educator–learner relationship between the first researcher and participants. Therefore, participants might have difficulties with raising concerns. In this study, the researcher tried to perform interviews with multiple participants in various situations and tried to ensure that participants did not feel conscious of the assessments of their training. Another limitation is transferability: this research was performed in only one rural hospital. To improve reliability, we used an audit trial, purposive sampling, and a long duration of collecting participants. Future studies should investigate effective reflection methods in other regions and international contexts. Additionally, the interview transcripts were coded by the first author, which could affect this study’s credibility. To improve the quality of research, the second author reviewed the process of coding, concepts, and themes through theoretical triangulation.

## 5. Conclusions

For bedside teaching in rural family medicine education to be effective, it needs to accommodate different learners, and facilitate near-peer and self-directed learning, adjust educators’ involvement based on cognitive apprenticeship, and encourage interprofessional collaboration with nurses. Interpersonal collaboration between rural family medicine teachers, learners, and nurses in bedside teaching may improve the quality of patient care based on the understanding of patients and other medical staff in hospitals.

## Figures and Tables

**Figure 1 ijerph-19-06807-f001:**
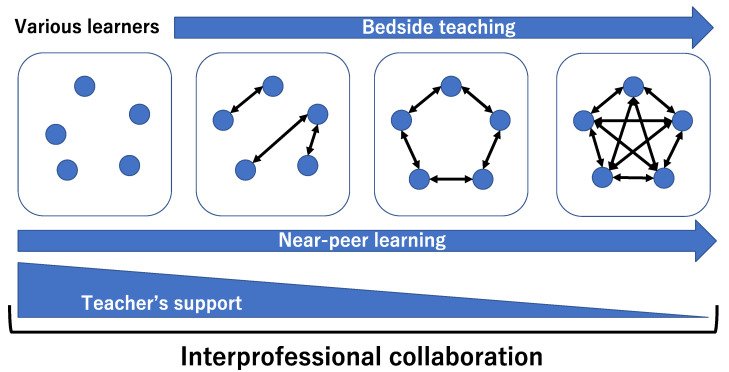
Framework of bedside teaching in rural family medicine education.

## Data Availability

The datasets used and/or analyzed during the current study may be obtained from the corresponding author upon reasonable request.

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
