# Peer review of "Bedside Teaching in Rural Family Medicine Education in Japan"

_ijerph, 2022, doi:10.3390/ijerph19116807_

Round 1
Reviewer 1 Report
Dear Author,
I have reviewed your manuscript and I think is interesting for the readers of this journal.
I suggest to introduce these points to improve the quality of the manuscript:
- Specify what type of Thematic Analysis you use, quoting authors and their models or techniques.
- To improve the relationship between education and technology from a qualitative perspective.
Author Response
Responses to the reviewers’ comments
Thank you very much for reviewing our manuscript and providing suggestions for its improvement. We have provided point-by-point responses to the reviewers’ comments; our revisions are indicated in red font here and in the document. We hope that the revised manuscript meets the journal’s requirements and can now be considered for publication.
Dear Author,
I have reviewed your manuscript and I think is interesting for the readers of this journal.
I suggest to introduce these points to improve the quality of the manuscript:
- Specify what type of Thematic Analysis you use, quoting authors and their models or techniques.
Response:
Thank you for the productive suggestions. We agree with the suggestions and added an explanation of inductive thematic analysis in the abstract and method section (Lines 119 to 125).
- To improve the relationship between education and technology from a qualitative perspective.
Response:
Thank you for the productive suggestions. We agree with the suggestions and added discussion of the relationship between the educational method of bedside teaching and technology such as ICT devices with respect to ethical considerations (Lines 294 to 301).
Reviewer 2 Report
It is a well-written manuscript. I wonder, though, if you would like to define 'bedside teaching' in its most simple way. You did mention what it is about and discussed its importance, especially in rural areas. However, for someone who may not know about that concept, it may be a little hard to fully understand what the concept is about.
Also, you are dealing with rural areas (which is great you decided to do that). So, how is 'culture" related to the concept? I am curious because people in rural areas may be very strong in terms of traditions, customs, etc.
Author Response
Responses to the reviewers’ comments
Thank you very much for reviewing our manuscript and providing suggestions for its improvement. We have provided point-by-point responses to the reviewers’ comments; our revisions are indicated in red font here and in the document. We hope that the revised manuscript meets the journal’s requirements and can now be considered for publication.
It is a well-written manuscript. I wonder, though, if you would like to define 'bedside teaching' in its most simple way. You did mention what it is about and discussed its importance, especially in rural areas. However, for someone who may not know about that concept, it may be a little hard to fully understand what the concept is about.
Response:
Thank you for the productive suggestions. We agree with the suggestions and added the relevant background concepts in the background section.
Regarding family medicine education, we added an explanation in the first paragraph in the background, as follows,
“Family medicine deals with many kinds of medical issues faced by patients in inpatient and outpatient departments. Physicians who specialize in family medicine are called family physicians and work in various clinical settings such as hospitals, clinics, and public health centers. Family physicians, thus, have to learn ways to solve various healthcare issues in their training [1]. In family medicine education, various educational methods are used, such as case-based learning, problem-based learning, and on-the-job training. One type of on-the-job training is bedside teaching [2].(Lines 30 to 36).
Regarding bedside teaching, we added an explanation of family medicine education in the first paragraph of the background section:
“One type of on-the-job training is bedside teaching [2].
“Bedside teaching is an on-the-job training performed in inpatient clinical wards to teach medical knowledge, skills, and professionalism. Usually, medical teachers and students go to patients’ bedside and examine the patients together. Medical teachers observe their students’ performance and give them feedback in front of patients or outside patients’ rooms [2,3]. Bedside teaching can be performed in various situations in clinical contexts, such as morning rounds, formal educational rounds, and attending rounds, to assess learners’ management of patients [3,4], each of which can improve learning among medical students. Nowadays, the prevalence of bedside teaching has declined because of computer-based working and dependence on laboratory tests for the management of patient conditions [5,6]. Nevertheless, to sustain effective, high-quality education, bedside teaching should be encouraged and improved [7].” (Lines 36 to 47)
Furthermore, we have revised the discussion to add in-depth explanations of near-peer learning, self-directed learning, and interprofessional collaboration (Lines 277 to 293).
Also, you are dealing with rural areas (which is great you decided to do that). So, how is 'culture" related to the concept? I am curious because people in rural areas may be very strong in terms of traditions, customs, etc.
Response:
Thank you for the productive suggestions. We agree with the suggestions and added rural cultural issues in the background section, as follows,
“Furthermore, certain characteristic features of rural contexts can affect family medicine education. Patients in rural hospitals may not be used to being examined by medical residents and teachers, because of the lack of culture of medical education [9]. Other medical professionals, such as nurses, in rural hospitals also may not have experienced educational situations where medical students and teachers discuss patients’ condition in medical wards [14,15]. Therefore, the application of bedside teaching in rural family medicine education can have challenges and difficulties. At present, there is a lack of evidence regarding the most effective ways of conducting bedside teaching in rural contexts. Bedside teaching is nonetheless necessary for effective rural family medicine education. So, this study’s purpose was to construct an effective framework for application of bedside teaching in rural family medicine education.” (Lines 56 to 66).
Reviewer 3 Report
Introduction:
There is lack of information regarding the rural japan. What type of specific issue rural Japanese face. How the bedside teaching is related and contributing to it. Kindly include those information under introduction.
Also, give brief idea about the rural family medicine education curriculum for the international readers.
There is no research gap mentioned.
What framework the authors refer to? is it based any theoritical model?
There is no specific objective provided.
Material and methods:
Study participants information is not clear. Kindly add a sub heading 'study sample' and provide information under it.
Under ethnography, whom do the authors refer as participants?
There is interchange of words as author and researcher. Kindly use either one to denote.
Nurses were part of this study? if yes, it is not mentioned in the study participants.
What was the sampling method used in this study?
There is no information on trustworthiness in qualitative research. Kindly include it.
Was triangulation done in this research?
Results:
The participants' statement should be italic.
Kindly include the demographic profile of the participants in a table format.
Discussion:
It is poorly written. The authors should describe, analyze, and interpret their findings. not produce information from other literature.
Author Response
Responses to the reviewers’ comments
Thank you very much for reviewing our manuscript and providing suggestions for its improvement. We have provided point-by-point responses to the reviewers’ comments; our revisions are indicated in red font here and in the document. We hope that the revised manuscript meets the journal’s requirements and can now be considered for publication.
Introduction:
There is lack of information regarding the rural japan. What type of specific issue rural Japanese face. How the bedside teaching is related and contributing to it. Kindly include those information under introduction.
Response:
Thank you for the productive suggestions. We agree with the suggestions and added these background concepts in the background section.
First, we added the explanation of family medicine education in the first paragraph in the background:
“Family medicine deals with many kinds of medical issues faced by patients in inpatient and outpatient departments. Physicians who specialize in family medicine are called family physicians and work in various clinical settings such as hospitals, clinics, and public health centers. Family physicians, thus, have to learn ways to solve various healthcare issues in their training [1]. In family medicine education, various educational methods are used, such as case-based learning, problem-based learning, and on-the-job training. One type of on-the-job training is bedside teaching.” (Lines 30 to 36).
Regarding bedside teaching, we added the following:
“One type of on-the-job training is bedside teaching [2].
“Bedside teaching is an on-the-job training performed in inpatient clinical wards to teach medical knowledge, skills, and professionalism. Usually, medical teachers and students go to patients’ bedside and examine the patients together. Medical teachers observe their students’ performance students and give them feedback in front of patients or outside of patients’ rooms [2,3]. Bedside teaching can be performed in various situations in clinical contexts, such as morning rounds, formal educational rounds, and attending rounds, to assess learners’ management of patients [3,4], each of which can improve learning among medical stu-dents. Nowadays, the prevalence of bedside teaching has declined because of computer-based working and dependence on laboratory tests for the management of patient conditions [5,6]. Nevertheless, to sustain effective, high-quality education, bedside teaching should be encouraged and improved [7].” (Lines 36 to 47)
Regarding bedside teaching, we added an explanation of rural contexts in the first paragraph in the background as following,
“Furthermore, certain characteristic features of rural contexts can affect family medicine education. Patients in rural hospitals may not be used to being examined by medical residents and teachers, because of the lack of culture of medical education [9]. Other medical professionals, such as nurses, in rural hospitals also may not have experienced educational situations where medical students and teachers discuss patients’ condition in medical wards [14,15]. Therefore, the application of bedside teaching in rural family medicine education can have challenges and difficulties. At present, there is a lack of evidence regarding the most effective ways of conducting bedside teaching in rural contexts. Bedside teaching is nonetheless necessary for effective rural family medicine education. So, this study’s purpose was to construct an effective framework for application of bedside teaching in rural family medicine education.” (Lines 56 to 66).
Also, give brief idea about the rural family medicine education curriculum for the international readers.
Response:
Thank you for the productive suggestions. We agree with the suggestions, and added contextual discussion of rural family medicine education in the method section as follows:
“The hospital followed a rural family medicine education curriculum, which included three family medicine teachers. Under this curriculum, residents experienced various clinical situations in the treatment of their patients. In their first year, residents worked at the Unnan City Hospital and treated typical diseases in both inpatient and outpatient situations. In the following year, they worked at a rural clinic (Kakeya Clinic) for six months to learn home care and community-oriented primary care. To broaden their scope of practice in internal medicine, pediatrics, and emergency medicine, they also worked at a general or community hospital for 18 months.
Each clinical setting included a medical teacher. The curriculum could be utilized to educate a maximum of three residents per teacher simultaneously. One resident in the years 2018 and 2019 and three in the years 2020 and 2021 engaged in the curriculum.
Medical students and junior residents had received rural family medicine education at medical universities and tertiary hospitals. They trained in family medicine at the rural hospital for two weeks to a month with medical teachers and family medicine residents. It was mandatory for them to work at the rural hospital, as part of their university or hospital curriculum. Each year, the rural hospital accommodates 40 to 50 medical students and junior residents for training.” (Lines 79 to 95)
There is no research gap mentioned. What framework the authors refer to? is it based any theoritical model? There is no specific objective provided.
Response:
Thank you for the productive suggestions. We agree with the suggestions and added material on the theoretical framework, evidence gap, and purpose of this research accordingly, as follows:
“In particular, bedside teaching is useful in the rural context, because rural family medicine deals with the broad range of biopsychosocial problems among older patients. Since rural community hospitals lack sufficient medical professionals devoted to full-time care, family medicine educators often treat patients in these rural communities as well as providing education [8,9,10,11]. When they become too busy with clinical practice, they may not be able to devote enough time to residents [12,13].
Furthermore, certain characteristic features of rural contexts can affect family medicine education. Patients in rural hospitals may not be used to being examined by medical residents and teachers, because of the lack of culture of medical education [9]. Other medical professionals, such as nurses, in rural hospitals also may not have experienced educational situations where medical students and teachers discuss patients’ condition in medical wards [14,15]. Therefore, the application of bedside teaching in rural family medicine education can have challenges and difficulties. At present, there is a lack of evidence regarding the most effective ways of conducting bedside teaching in rural contexts. Bedside teaching is nonetheless necessary for effective rural family medicine education. So, this study’s purpose was to construct an effective framework for application of bedside teaching in rural family medicine education.” (Lines 48 to 66)
Material and methods:
Study participants information is not clear. Kindly add a sub heading 'study sample' and provide information under it.
Response:
Thank you for the productive suggestions. We agree with the suggestions and added a subsection ‘participants’ in the methods section (Lines 97 to 102).
Under ethnography, whom do the authors refer as participants?
Response:
Thank you for the generative question. We added explanation of the participants in the ethnography. In this research, the first researcher observed the interactions among medical students, residents, nurses, and patients involved in bedside teaching. We have added the following description:
“The first researcher conducted ethnography and semi-structured interviews with the participants: medical students, residents, and teachers involved in the education and their patients. In the ethnography, participants were medical students, residents, nurses and patients in bedside teaching situations. This researcher’s specialties are family medicine, medical education, and public health. The researcher worked in all hospital wards and observed the interactions among medical students, residents, teachers, and nurses on bedsides, and took field notes during this process.” (Lines 104 to 110).
There is interchange of words as author and researcher. Kindly use either one to denote.
Response:
Thank you for the productive suggestions. We agree with the suggestions, and changed the word “author” to “researcher” across the whole manuscript.
Nurses were part of this study? if yes, it is not mentioned in the study participants.
Response:
Thank you for raising this point. We agree, and added description of the participants, including nurses (Lines 99).
What was the sampling method used in this study?
Response:
Thank you for this question. We added an explanation of the purposive sampling method in the participants section (Lines 101 to 102).
There is no information on trustworthiness in qualitative research. Kindly include it. Was triangulation done in this research?
Response:
Thank you for the productive suggestion. We agree, and added a description of credibility, triangulation, audit trial, and theoretical saturation contributing to trustworthiness in the analysis section:
“Inductive thematic analysis was used. After reading the field notes and interview transcriptions in depth, the first researcher coded the contents and developed codebooks based on repeated reading of the research materials, as initial coding for reliability. This study used process and concept coding; the researcher, thus, induced, merged, deleted, and refined the concepts and themes by going back and forth between the research materials and the initial coding. For triangulation, the concepts and themes were discussed among the first and second researchers. The interview data were collected iteratively during the research period each month from different participants for theoretical saturation. The tentative analysis results were also shared with the participants for audit trial. Finally, the theory was discussed by both researchers, who ultimately reached agreement on the final themes.” (Lines 119 to 129).
Results:
The participants' statement should be italic.
Response:
Thank you for the productive suggestion. We agree, and have changed the participants’ statements to italics throughout.
Kindly include the demographic profile of the participants in a table format.
Response:
Thank you for the productive suggestion. We added a brief description of the participants in the results section. Because of lack of data of the background of participants, we briefly describe the background data in a few sentences (Lines 132 to 133).
Discussion:
It is poorly written. The authors should describe, analyze, and interpret their findings. not produce information from other literature.
Response:
Thank you for the productive suggestion. We agree, and have added our findings, with their interpretation and suggestions for future studies. To do so, we have revised the whole discussion (Lines 270 to 349).
Round 2
Reviewer 1 Report
I think the article can be published.
Reviewer 3 Report
Comments are addressed